# Numerical Modeling of Space–Time Characteristics of Plasma Initialization in a Secondary Arc

**Jinsong Li \*, Hua Yu, Min Jiang, Hong Liu and Guanliang Li**

Equipment State Analysis Center, State Grid Shanxi Electric Power Research Institute, Taiyuan 030001, China; yuhua16885@163.com (H.Y.); jigmi@163.com (M.J.); 18935121293@189.cn (H.L.); 15386812717@163.com (G.L.)

\* Correspondence: ljskssss@sina.com

**Abstract:** A numerical model based on the finite element simulation software COMSOL was developed to investigate the secondary arc that can limit the success of single-phase auto-reclosure solutions to the single-phase-to-ground fault. Partial differential equations accounting for variation of densities of charge particles (electrons, positive and negative ions) were coupled with Poisson's equation to consider the effects of space and surface charges on the electric field. An experiment platform was established to verify the numerical model. The brightness distribution of the experimental short-circuit arc was basically consistent with the predicted distribution of electron density, demonstrating that the simulation was effective. Furthermore, the model was used to assess the particle density distribution, electric field variation, and time dependence of ion reactions during the short-circuit discharge. Results showed that the ion concentration was higher than the initial level after the short-circuit discharge, which is an important reason for inducing the subsequent secondary arc. The intensity of the spatial electric field was obviously affected by the high-voltage electrode at the end regions, and the intermediate region was mainly affected by the particle reaction. The time correspondence between the detachment reaction and the ion source generated in the short-circuit discharge process was basically consistent, and the detachment reactions were mainly concentrated in the middle area and near the negative electrode. The research elucidates the relevant plasma process of the secondary arc and will contribute to the suppression of it.

**Keywords:** secondary arc; short-circuit discharge; numerical modeling; plasma discharge

## 1. Introduction

Because most of the faults on ultra-high voltage (UHV) and extra-high voltage (EHV) transmission lines are typically impermanent single-phase-to-ground faults, single-phase auto-reclosure (SPAR) can eliminate most of their potential effects. If a single-phase-to-ground fault occurs along the high-voltage transmission lines, a short-circuit arc discharge of large current will be incurred at the fault point. After the fault phase is switched off, the short-circuit arc will be extinguished. However, due to the electromagnetic (EM) coupling between transmission lines, a secondary arc discharge of small current will continue through the same arc path at the fault point. Therefore, the timely extinction of the secondary arc caused by the single-phase ground fault is important for the success of single-phase reclosing. To ensure the safe operation of the power transmission lines and enhance the stability of the power system, a method that enables the self-extinction of the secondary arc is urgently needed to be found [1–3].

Although physical experiments can be performed to study the secondary arc characteristics directly, such experiments are restricted by environmental conditions, require large investments, and are insufficiently flexible [4–7]. In a circuit simulation model, the fault arc is very often represented by a time-varying resistor, and it is described via nonlinear differential equations [8–10]. The arc chain model, in which arc movement is modelled with consideration of the electromagnetic force,

thermal buoyancy, wind load force, and air resistance, has been adopted by many researchers to obtain the velocity equation of arc movement through the force analysis of each arc element [11–14]. However, these models do not consider the plasma produced by the discharge.

Numerical simulations are particularly suitable for analyzing and optimizing the complex plasma processes created by the air discharge, and these plasma processes can be further elucidated by comparing predictions from numerical simulations with experimental observations. Various modelling approaches have been adopted by experts, including analytical models, fluid models, non-equilibrium Boltzmann equations, Monte Carlo simulations, and particle-in-cell models [15]. Hybrid models that combine some of these models are also used [16], as further detailed in [17]. In particular, hydrodynamic fluid models have been shown to offer the advantages of efficiency, accuracy, and comprehensiveness, and have most often been employed [18,19]. However, due to extensive calculations and complex external conditions, the research has been focused on short gap discharges, such as corona discharge, dielectric barrier discharge, and other fields.

Although the secondary arc has been thoroughly studied experimentally and the arc chain model has been applied to analyze the movement of the secondary arc, few studies have been conducted on the plasma process of the secondary arc. Therefore, the aim of this study was to develop a finite element model for the secondary arc, using the simulation software COMSOL and focusing on the initial stage, and to elucidate the relevant plasma processes by comparing predictions from numerical simulations with experimental observations.

## 2. Model Description

### 2.1. Governing Equations

The most widely used formulation of a streamer propagation model in air is based on the drift-diffusion hydrodynamic approach, which considers variations in the densities of electrons and two generic types of ions (positive and negative) in space and time (see, e.g., [1–6]). This approach results in the following three partial differential convection–diffusion equations, which also account for rates of the physical processes leading to the generation and loss of charged species:

$$\frac{\partial N_e}{\partial t} + \nabla \cdot (-D_e \nabla N_e) + \beta_e \cdot \nabla N_e = f_e, \tag{1}$$

$$\frac{\partial N_p}{\partial t} + \nabla \cdot (-D_p \nabla N_p) + \beta_p \cdot \nabla N_p = f_p, \tag{2}$$

$$\frac{\partial N_n}{\partial t} + \nabla \cdot (-D_n \nabla N_n) + \beta_n \cdot \nabla N_n = f_n. \tag{3}$$

here, the subscripts e, p, and n indicate electrons and positive and negative ions, respectively; $N$ is the density, in $m^{-3}$; $D$ is the diffusion coefficient, in $m^2/s$; $f$ is the net rate of the generation and loss processes, in $m^{-3}s^{-1}$; and $t$ represents time, in s. The main processes usually considered in Equations (1) and (3) are represented by their rates: Electron impact ionization, $f_{ion} = \alpha N_e \mu_e E$; attachment of electrons to electronegative molecules ($CO_2$, $H_2O$, $O_2$, etc.) present in air, $f_{att} = \eta N_e \mu_e E$; detachment of electrons from negative ions, $f_{det} = k_{det} N_e N_n$; electron-ion recombination, $f_{ep} = \beta_{ep} N_e N_p$; recombination of positive and negative ions, $f_{pn} = \beta_{pn} N_p N_n$; and natural background ionization, $f_0$. In the expressions above, $\alpha$ is Townsend's ionization coefficient, in $m^{-1}$; $\mu$ is the mobility, in $m^2/Vs$; $E$ is the electric field strength, in V/m; $\eta$ is the attachment coefficient, in $m^{-1}$; $k_{det}$ is the detachment coefficient, in $m^3/s$; and $\beta$ is each respective recombination coefficient, in $m^3/s$. Hence, the net rates for different charged species are:

$$f_e = f_{ion} + f_{det} + f_0 - f_{att} - f_{ep}, \tag{4}$$

$$f_p = f_{ion} + f_0 - f_{pn} - f_{ep}, \tag{5}$$

$$f_n = f_{att} - f_{det} - f_{pn}. \tag{6}$$

Equations (1) and (3) must be complemented by Poisson's equation for electric potential $V$. The solution provides the electric field distributions affected by the space charge, which are needed to obtain the kinetic coefficients and the rates of individual processes:

$$\nabla(-\varepsilon_0\varepsilon_r\nabla V) = e(N_p - N_e - N_n),\tag{7}$$

$$-\nabla V = \boldsymbol{E}.\tag{8}$$

here, $e$ is the elementary charge, $\varepsilon_0$ is the vacuum permittivity, and $\varepsilon_r$ is the dielectric constant of the material (unity for air). Equations (1) and (8), with boundary and initial conditions specific to the problem, form a self-consistent model that must be solved numerically because of the strongly non-linear nature of the model.

Parameters and rate coefficients in the hydrodynamic models should be obtained from a solution of Boltzmann's equation. Local field approximations are assumed, i.e., gas properties such as drift velocities and the collisional ionization coefficient are functions only of $E/N$, where $E$ is the field amplitude and $N$ the gas number density. In this study, transport coefficients needed for simulations of discharges are obtained from [20], and the results are verified by comparing them with those obtained using the popular Boltzmann equation solver. The transport coefficients are also compared with the experimental results for air, and the achieved agreement confirms the validity of the parameters utilized [21].

The rate and kinetic coefficients used in the model are provided in Table 1. The dependencies of the ionization and attachment coefficients on the field strength are reproduced in Figure 1. The dependencies of the electron drift velocity and diffusion coefficient are approximated as $w_e = 3.2 \times 10^3 \times (E/N)^{0.8}$ m/s and $D_e = 7 \times 10^{-2} + 8 \times (E/N)^{0.8}$ m$^2$/s, respectively. The parameters are selected based on analysis of the literature and slightly adjusted according to the convergence of the model.

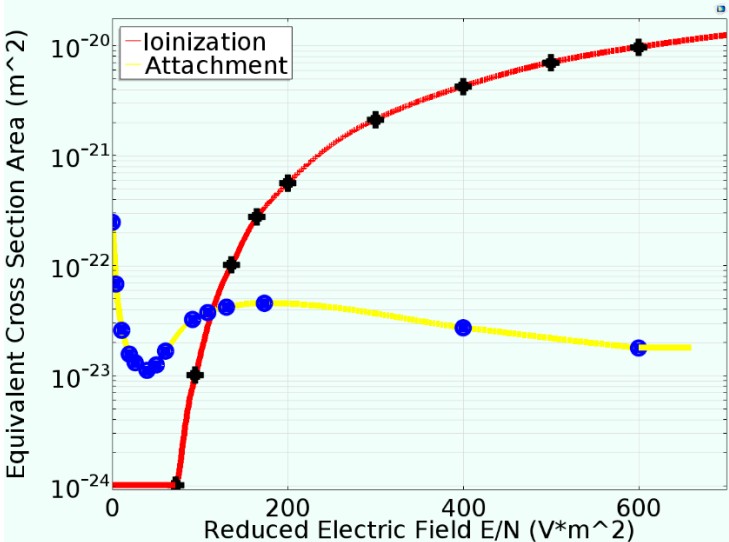

**Figure 1.** Ionization and electron attachment coefficients as functions of the reduced field $E/N$.

**Table 1.** Input parameters for the model.

| Transport Parameters | Expression | Description |
|---|---|---|
| $\mu_p$ (m$^2$/Vs) | $2.0 \times 10^{-6}$ | Positive ion mobility |
| $D_p$ (m$^2$/s) | 5.05 | Positive ion diffusivity |
| $\mu_n$ (m$^2$/Vs) | $2.2 \times 10^{-6}$ | Negative ion mobility |
| $D_n$ (m$^2$/s) | 5.56 | Negative ion diffusivity |
| $\beta_{ep}$ (m$^3$/s) | $5.0 \times 10^{-14}$ | Electron-positive ions recombination rate |
| $\beta_{pn}$ (m$^3$/s) | $2.07 \times 10^{-13}$ | Positive-negative ions recombination rate |
| $f_0$ (1/m$^3$s) | $1.7 \times 10^9$ | Natural background ionization source item |
| $k_{det}$ (m$^3$/s) | $1 \times 10^{-18}$ | Electron detachment coefficient from negative ions |

## 2.2. Computational Domain and Meshing

The geometric model of the secondary arc simulation conducted in this study is shown in Figure 2a. Circular symmetry was exploited in constructing the geometric configuration. The whole computation domain was 1.62 m high and 0.4 m wide. The insulator string was 1 m long (i.e., the distance between the top and bottom electrode), and the radius of the center column of the insulator was 0.025 m. The ambient temperature T = 288.15 K (15 °C), and the background pressure was 1 atm (1 atmosphere = $1.01325 \times 10^5$ Pa).

The above structures were meshed by free triangles, as shown in Figure 2b. The model grid contained 10,821 triangles, with a maximum grid size of 0.06 m and a minimum grid size of 0.01 m. Near the electrodes and the ignition line, the charge density and its variation are particularly strong, which demands a very fine spatial mesh, whereas the rest of the discharge space rarely exhibits the steep gradients associated with the electrodes. As can be seen from Figure 2b, at the electrode region and ignition line a very fine resolution was employed to resolve the steep gradients, as required, but away from the axis of symmetry, a very coarse mesh was used as the charge density does not vary greatly there.

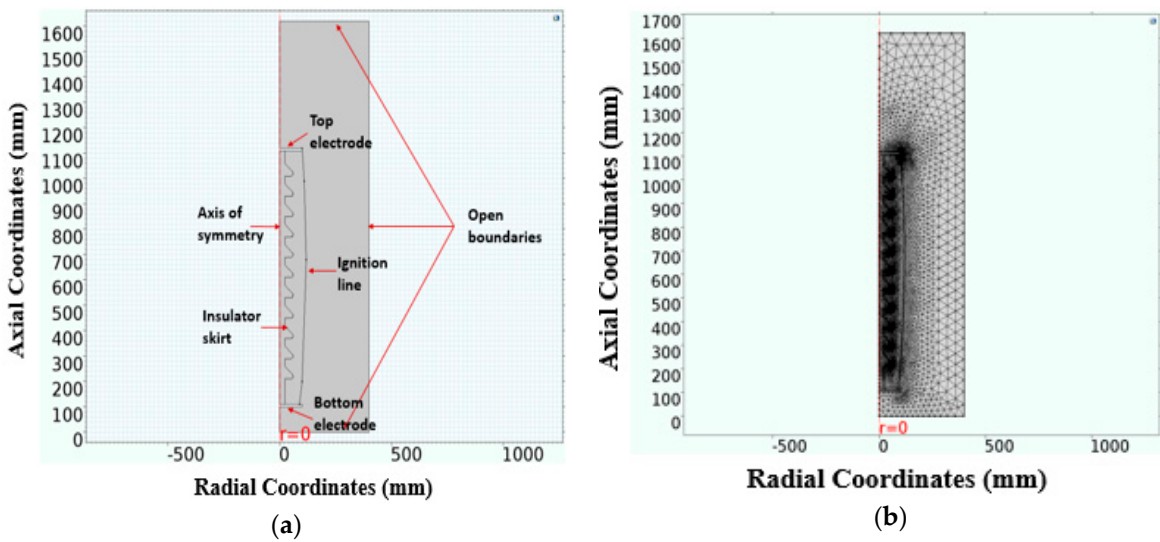

**Figure 2.** The geometric model and meshing result of the secondary arc simulation: (**a**) The geometric model; (**b**) meshing result.

## 2.3. Numerical Modelling of the Short-Circuit Arc

In the experiment study, a fuse was usually adopted to ignite the arc discharge and simulate the short circuit, and then the secondary arc was generated. For the purposes of this study, an ignition line was added to the simulation model, and the emission particle source (electron $g_{e2}$, positive ion $g_{p2}$, negative ion $g_{n2}$) was set on the ignition line to simulate the high-charge-density arc channel generated

by the short-circuit combustion. The particle sources were established with the following Gaussian pulse functions (Figure 3) [22]:

$$\begin{cases} g_{e2} = 1 \times 10^{13} \cdot gp1(t) \\ g_{p2} = 3 \times 10^{13} \cdot gp1(t) \\ g_{n2} = 2 \times 10^{13} \cdot gp1(t) \end{cases}, \tag{9}$$

where $gp1(t)$ is the Gaussian impulse function. The width of the impulse depends on the duration of short-circuit discharge (about 0.2 s). The amplitude of the impulse has positive correlation with the short-circuit arc current (1 kA in this study). The appropriate value of the impulse at the center position was calculated to be 0.08 with a standard deviation of 0.05.

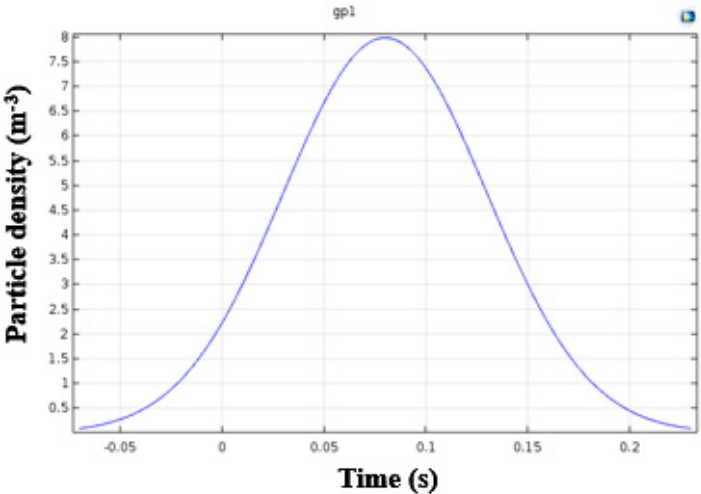

**Figure 3.** Gaussian impulse function.

## 2.4. Boundary and Initial Conditions

The boundary conditions adopted for the numerical simulation of the geometric structure shown in Figure 2a are summarized in Table 2. Because circular symmetry was adopted, the variation of three charged particles along the radial direction was zero. Considering the adsorption and recombination of negative ions by the positive electrode, the concentration of negative ions was set to zero, and the positive ion concentration was similarly set to zero on the surface of the negative electrode. On the top electrode (positive electrode), there were accumulative positive ions from ion reactions, and there were electrons and negative ions on the bottom electrode. The ignition line was loaded with the transient Gaussian pulse source according to Equation (9). These are described by the flux/source boundary condition. A zero-flux boundary was established to define the rest of the outer boundary of the calculation domain for the ion transport process [23]. The PARDISO transient solver embedded in COMSOL was utilized, and the time step of the solver was controlled by the BDF (Backward Differentiation Formulas) algorithm. The simulation time started the moment the short-circuit power was connected to 0 and ended at 0.2 s.

**Table 2.** Boundary conditions.

| Application Location | Convection and Diffusion $N_e$ | Convection and Diffusion $N_p$ | Convection and Diffusion $N_n$ |
|---|---|---|---|
| Axis of symmetry | $\frac{\partial N_e}{\partial r} = 0$ | $\frac{\partial N_p}{\partial r} = 0$ | $\frac{\partial N_n}{\partial r} = 0$ |
| Top electrode | $N_e = 0$ | $-\boldsymbol{n} \cdot (D_p \nabla N_p) = f_+$ | $N_n = 0$ |
| Bottom electrode | $-\boldsymbol{n} \cdot (D_e \nabla N_e) = f_-$ | $N_p = 0$ | $-\boldsymbol{n} \cdot (D_n \nabla N_n) = f'_-$ |
| Ignition line | $-\boldsymbol{n} \cdot (D_e \nabla N_e) = g_{e2}$ | $-\boldsymbol{n} \cdot (D_p \nabla N_p) = g_{p2}$ | $-\boldsymbol{n} \cdot (D_n \nabla N_n) = g_{n2}$ |
| Rest of the boundary | $-\boldsymbol{n} \cdot (D_e \nabla N_e) = 0$ | $-\boldsymbol{n} \cdot (D_p \nabla N_p) = 0$ | $-\boldsymbol{n} \cdot (D_n \nabla N_n) = 0$ |

Considering the sustainability of the ionic reaction, in its initial stage, three ion initial concentrations were set to $1 \times 10^{13}/\text{m}^3$, and 600 kV was loaded on the top electrode.

### 2.5. Experimental Platform

An equivalent single-phase experimental circuit was established according to the distributed parameter model for transmission lines, as shown in Figure 4a. Here, the inductance *L* establishes an inductive short-circuit starting current. In the experiment, this was 0.03688 H and the short-circuit current was 1 kA. The capacitance *C* represents the equivalent coupling capacitance between the faulty phase and healthy phases. This was 2.74 μF in this study. Different secondary arc currents were achieved by changing the value of the group capacitance *C* in the experiment. A voltage divider, current transformer, and oscilloscope were used to measure current and voltage waveforms in real time, and two high-speed cameras were used to record the entire discharge process at 4000 fps.

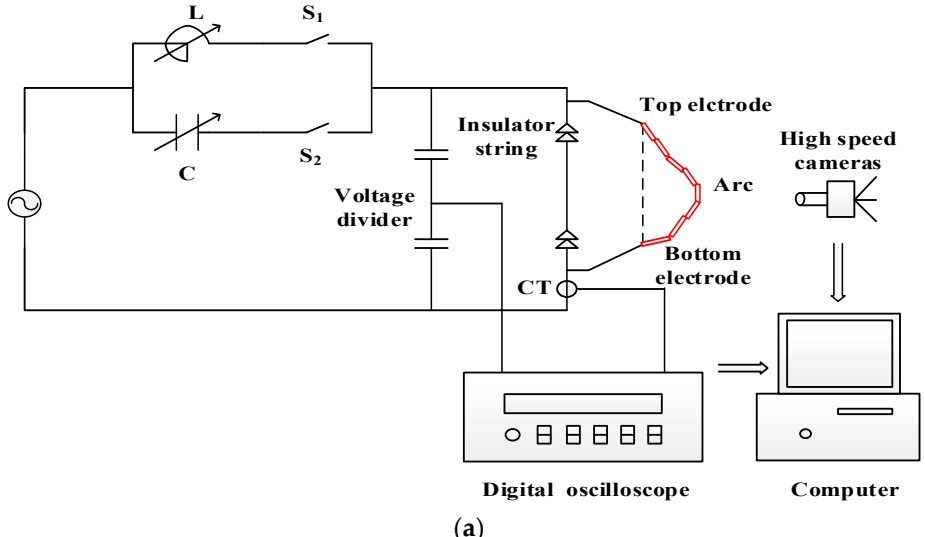

(**a**)

**Figure 4.** *Cont.*

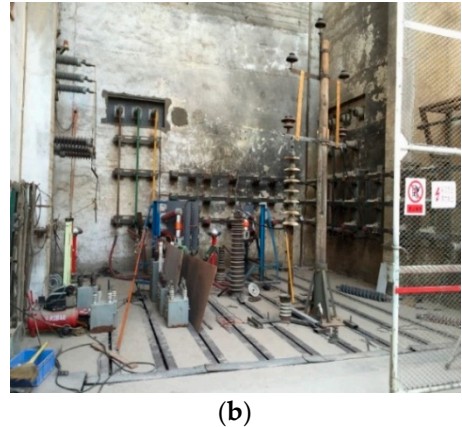
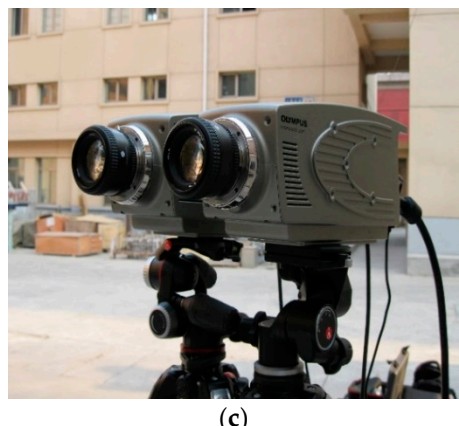

(**b**)                                                    (**c**)

**Figure 4.** Experimental platform for secondary arc reproduction: (**a**) Experimental circuit; (**b**) experiment field; (**c**) high-speed cameras.

Firstly, the circuit breaker S1 was closed to simulate an inductive arc starting current. Under the action of large current, the ignition line gasified to form an arc channel. After 0.1 s, the circuit breaker S2 closed and S1 quickly opened to simulate the secondary arc. The secondary arc experiment was completed in the high-current test station of China Electric Power Research Institute, shown in Figure 4b,c.

## 3. Results and Discussion

### 3.1. Experiment Verification

Figure 5 shows images of the short-circuit arc. Under the action of large short-circuit current, the ignition line rapidly fused and strongly ionized the air around the insulator, forming a bright arc plasma channel. Over time, the arc channel continued to spread, even after the breaker cut off the power (0.1 s) due to the powerful thermal effect. It was not until 0.15 s that the attenuation phenomena such as arc passage narrowing and brightness weakening appeared obviously. Due to the strong ionization and thermal effect of the short-circuit current, there was no zero-crossing stage, which is typical in an ac arc. Due to the huge current value of the short-circuit arc and the strong ionization of the surrounding air, the movement of the arc passage was mainly radial diffusion. The force of each part of the arc passage was mainly internal electromagnetic force, and external force had little influence. Thus, the arc was warped internally with no significant upward or left–right drift.

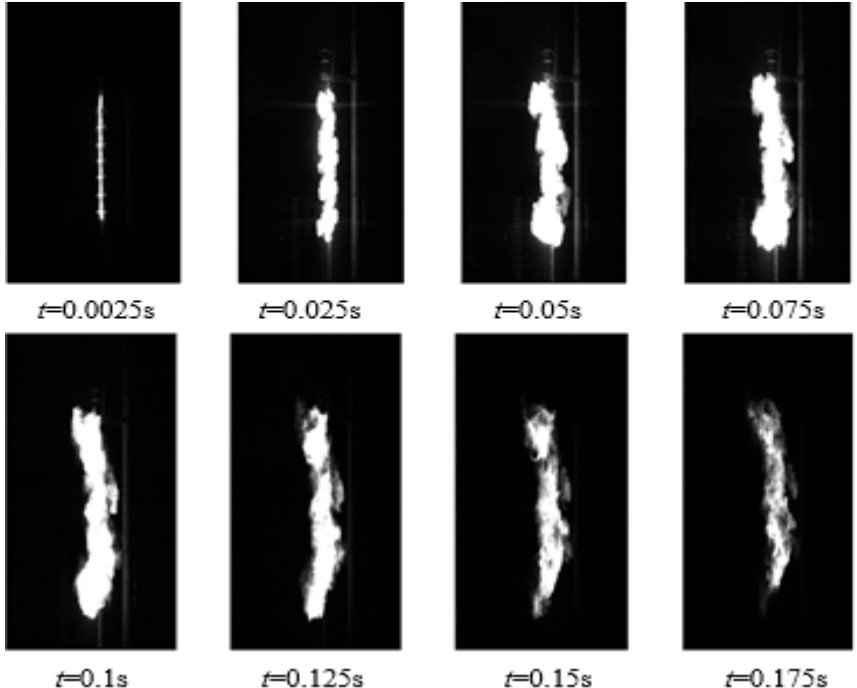

**Figure 5.** Short-circuit arc images at different times.

Figure 6 shows electron density distribution at different times during the short-circuit arc. At 0.0025 s, the short-circuit current started to melt the ignition line, exhibiting a luminous effect. During 0.025–0.15 s, the electrons produced by strong ionization concentrated near the ignition line, and gradually spread to the surrounding space under the action of the electric field migration and the convection diffusion of particles. After 0.15 s, the short-circuit arc decayed and the ionization region reduced. During this phase, the electron density distribution tended to return to the initial level.

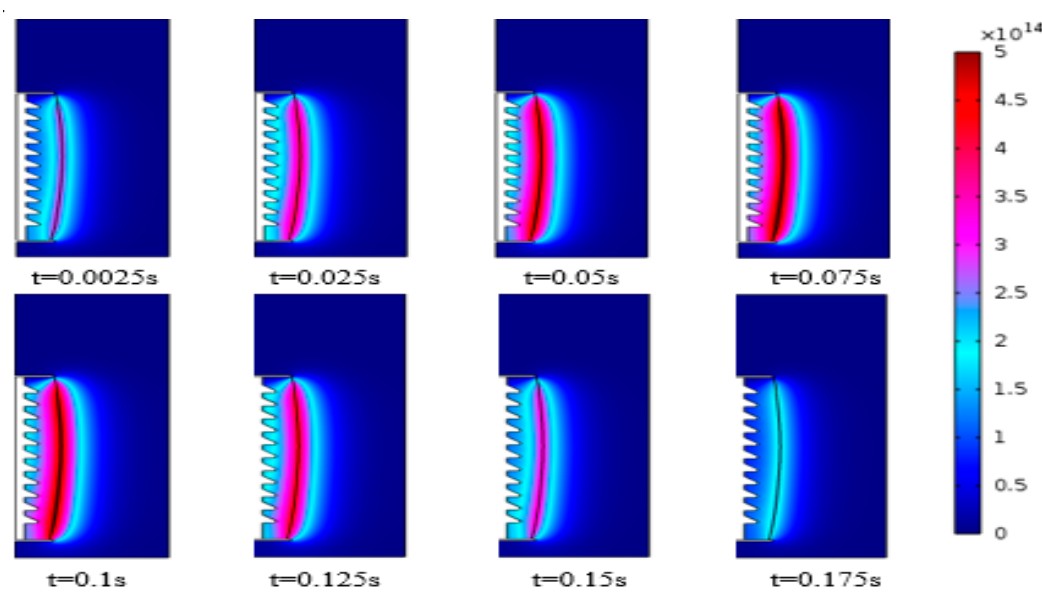

**Figure 6.** Electron density distribution at different times.

Luminescence is caused by the emission of photons when some equations in the plasma ionization reaction system transition from high to low energy levels. It is assumed that under the same environment, other ionization equations have the same reaction rate as the ionization reaction that emits photons. In this way, it can be considered to evaluate the ionization degree by observing

the luminescence intensity with cameras, and comparing the camera images with the simulated images to verify the consistency between the experiment and simulation. Comparing Figure 6 with Figure 5, it can be seen that the brightness distribution acquired with high-speed cameras of the experimental short-circuit arc was basically consistent with the predicted distribution of electron density, demonstrating that the simulation was effective and supporting the subsequent analysis of the plasma interior.

### 3.2. Particle Density Distribution and Development Law

Figure 7 plots the electron concentration distribution along the ignition line at six different times. The abscissa represents the arc length between the point on the ignition line and the 0 point on the negative electrode surface. The electron density slowly increased from the cathode region to the intermediate plasma region and remained constant until close to the anode region. After that, it sharply decreased and fixed to zero on the anode surface. Notice that the electron density did not increase significantly from the cathode arc root to the intermediate plasma region, which is markedly different from general streamer discharge. This was due to the strong joule heating effect of the short-circuit arc with large current, which caused strong ionization of the surrounding air, so there was little breakdown caused by electron collision ionization. Near the anode surface, the rapid drop was caused by the absorption of electrons by the anode. During the short-circuit arc phase, the peak electron concentration reached $5.72 \times 10^{14}$ m$^{-3}$, and the electron concentration was $1.3 \times 10^{14}$ m$^{-3}$ above what it was at the end of the simulation, which proves that the short-circuit discharge increased the concentration of space charge and provided necessary environmental conditions for the generation of a subsequent secondary arc.

Figure 8 shows the law of spatial negative ion density changing with time from the initial moment to 0.2 s on the surface of the negative pole (point 53) and the positive pole (point 58) as well as the middle point of the ignition line (point 60), which quantitatively reflects the gradual change of the concentration of transient particles in the short-circuit discharge process. It is not difficult to see that over time, the concentration of negative ions rose and then levelled off. When the ion reaction approached the end of simulation time, the ion concentration was higher than the initial level, which proves that the short-circuit discharge increased the spatial ion concentration and provided necessary environmental conditions for the subsequent secondary arc. Due to the difference of diffusion, convection, and adsorption coefficients between positive ions and negative ions, the changing curves of concentrations of positive ions and negative ions had slight differences despite showing the same trend.

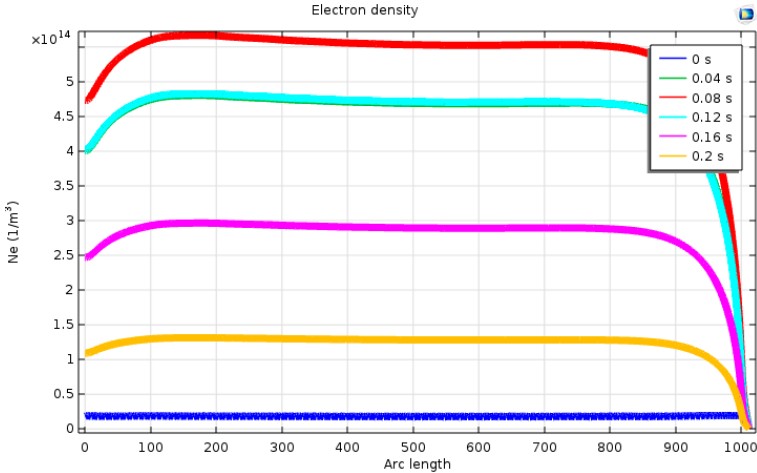

**Figure 7.** Electron density distribution along the ignition line at key time nodes during the discharge.

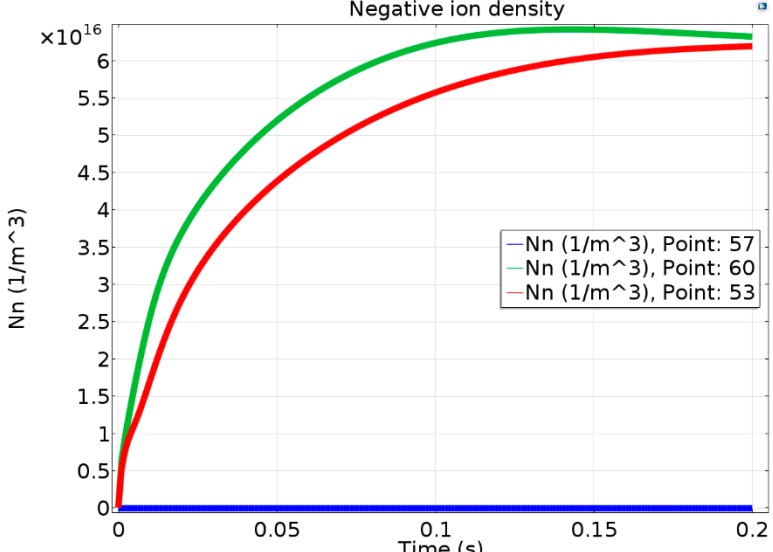

**Figure 8.** The negative ion density at the positive electrode (point 58), the negative electrode surface (point 53), and the central point (point 60) during the discharge.

### 3.3. Spatial Distribution of the Electric Field During Discharge

Figure 9 depicts the spatial electric field distribution in the initial stage of discharge, which was mainly caused by the high voltage applied by the electrode. In Figure 9, the intensity of the electric field is given by the cloud diagram, and the direction of the electric field is described by the red arrow. The electric field line started at the positive pole, crossed the whole space, and ended at the negative pole. Moreover, a large field intensity was generated at the maximum geometric curvature radius. This field intensity caused the point discharge.

Because the electric field generated by the ions was considerably different from the electric field generated by the electrodes, the field intensity effects at different times cannot be intuited from the cloud map, and are better represented by a one-dimensional graph. Figure 10 plots the electric field intensity as a function of the time in the middle region of the discharge space ($r = 120$ mm; $Z = 680$ mm, as an example). With increased discharge time, the electric field intensity at this spatial point showed an S-shaped upward trend. The essential reason for this rising trend is that a large number of ions are generated in the discharge process, and the electric field intensity generated by ions follows Gaussian electric field distribution, as described in Equations (7) and (8). Because the evaluated point was close to the short-circuit discharge area, the electric field tended to increase. When the discharge entered the later stage, the ions generated migrated under the action of the electric field and spread to other regions, causing the rate of the electric field intensity increase to attenuate gradually. The electric field intensity reaches the peak at about 0.2 s and then slowly decays. It takes 1 s usually for the extinction of the secondary arc, and if the simulation time is sufficiently long, the final electric field intensity could be predicted to return to its initial level [24].

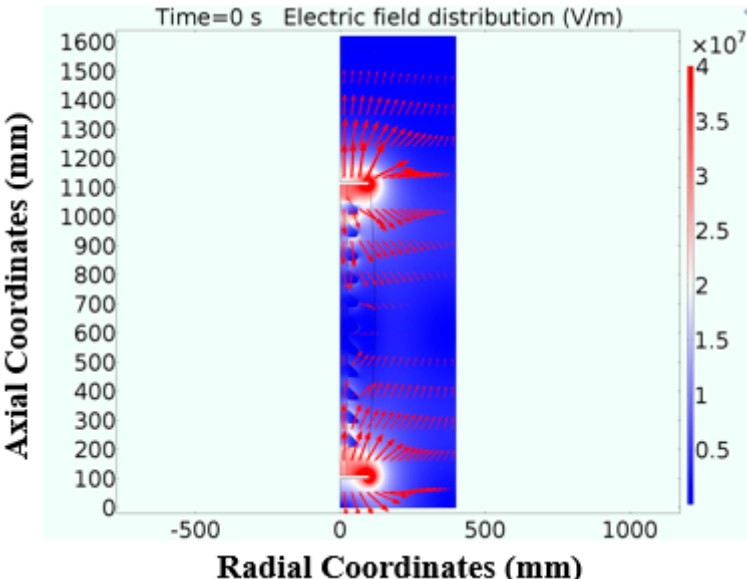

**Figure 9.** Spatial electric field distribution at the initial moment.

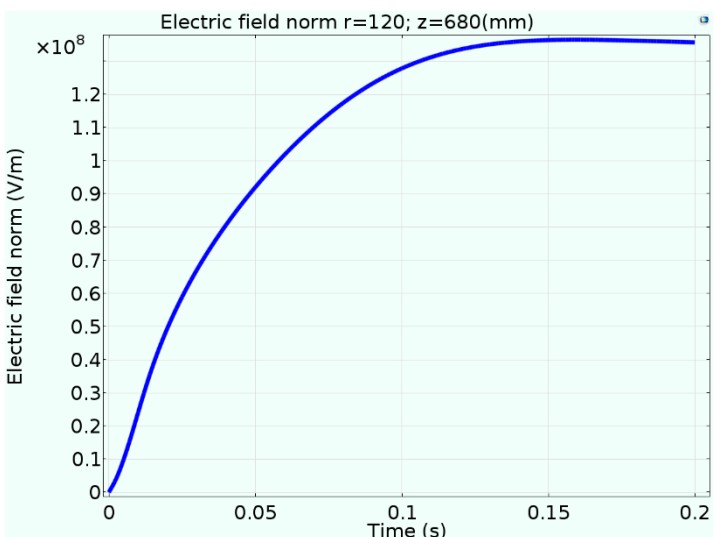

**Figure 10.** Electric field intensity over time at the midpoint of the discharge area.

To study the development of the electric field intensity over time, the transversal at $r = 100$ between the positive and negative electrodes was selected as an additional evaluation object. The development of the axial electric field intensity during the initial, peak, final, and intermediate stages of the discharge process was analyzed. Figure 11 shows that the electric field generated by the high voltage of the electrode was the strongest near the electrode and the lowest in the middle of the discharge region. Evidently, the contribution of ions to the spatial electric field was smaller than that of the high-voltage electrode; nonetheless, the former cannot be ignored. The contribution of ions to the spatial electric field was the largest in the intermediate discharge region, and the effect was small near the electrode.

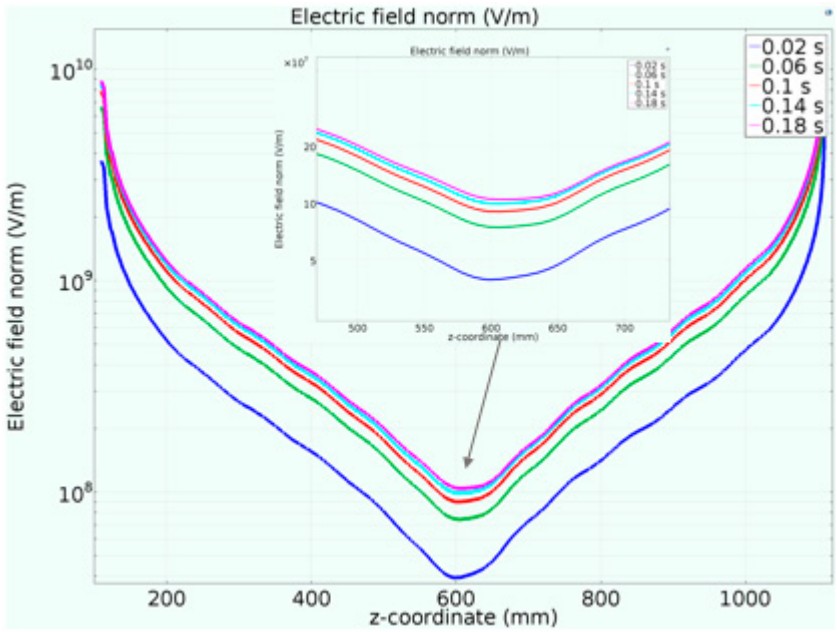

**Figure 11.** Development of axial electric field intensity over time at $r = 100$ mm.

### 3.4. Particle Reactions in Discharge Process

Figure 12 shows the time-dependent calculation results of the average detachment reaction rate in the discharge area. The time correspondences of the detachment reaction and ion source generated during the short-circuit discharge (Figure 3) is basically consistent: the detachment reaction rate increased sharply during the initial stage and peaked at the same time as the short-circuit discharge ion sources simulated by the Gaussian pulse. According to the formula $f_{\text{det}} = k_{\text{det}} N_e N_n$, the detachment of electrons from negative ions is influenced by both the concentration of electrons and the concentration of negative ions. This result occurs because numerous negative ions and electrons are rapidly generated in the short-circuit discharge. After the completion of the short-circuit discharge, the detachment reaction speed decreases gradually, unlike the rapid decay of ion sources concentration. That is because the recombination reaction process is influenced by the slow diffusion and migration of ions in space.

Considering the assumption of electrical neutrality during the initial stage of the discharge process, the detachment reaction rate in the initial stage was relatively uniform. Therefore, this section focuses on the spatial distribution of the detachment rate in the peak stages of the short-circuit discharge. On the surface plot Figure 13, one can observe the detachment reaction rate at the discharge peak was the highest near the ignition line and the lowest near the positive electrode. This is confirmed in Figure 14, which shows the detachment rate along the transversal at $r = 100$ between the positive and negative electrodes. As the bodies involved in the detachment reaction were negative ions and electrons, which were absorbed and neutralized near the positive electrode, the detachment reaction speed near the positive electrode was reduced. During the later stage of discharge, the negative ions and electrons generated by the short circuit tended to be uniform under the action of the electric field migration and diffusion. Therefore, the distribution of the detachment reaction rate gradually returned to the initial state.

Furthermore, the average recombination reaction (both electron-positive ions and positive-negative ions) rate was studied during the discharge. The results demonstrate that they are consistent with the trend of the detachment reaction rate, with differences only in magnitude. The influencing factors and mechanisms of each stage are also consistent with the detachment reaction, with differences caused only by the respective reaction coefficients. These reaction coefficients are affected by the ion collision cross sections and are selected for normal temperature and pressure conditions.

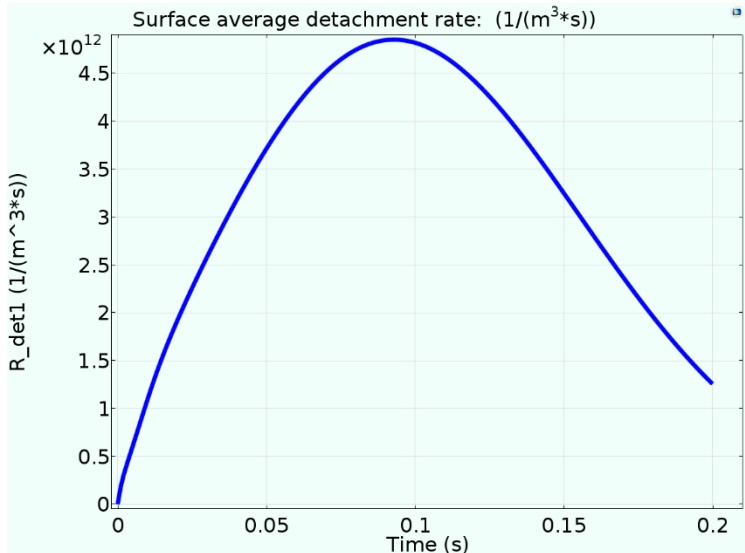

**Figure 12.** Time dependence of the average detachment reaction in the discharge space.

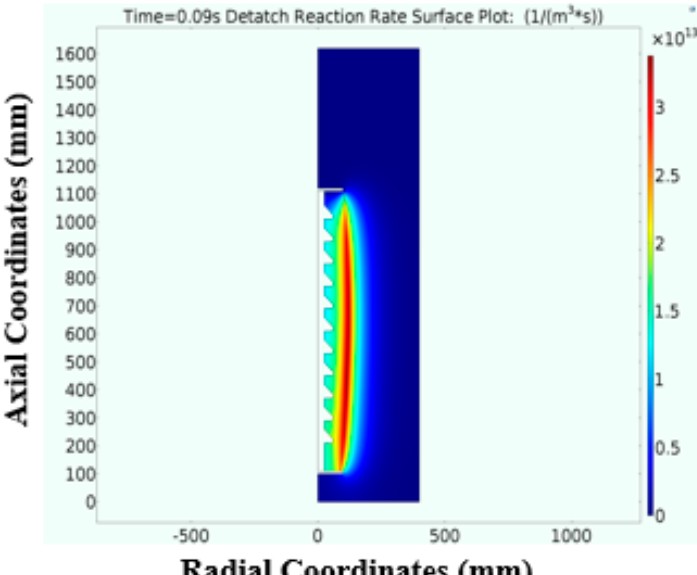

**Figure 13.** Detachment reaction rate surface plot at the discharge peak.

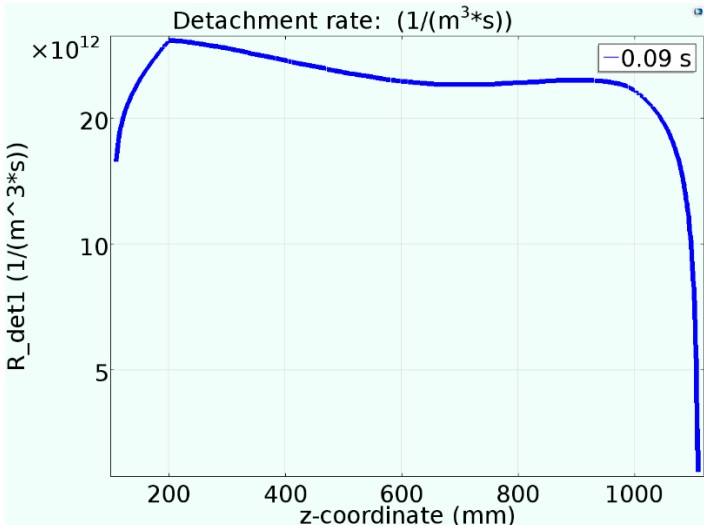

**Figure 14.** Detachment reaction rate at *r* = 100 mm at the peak stage of discharge.

## 4. Conclusions

In this study, COMSOL multi-physical field coupling analysis software was used to simulate the short-circuit discharge process during the initial stages of a secondary arc fault based on the relevant partial differential equations. The model particularly focused on the complex plasma created by the short-circuit discharge (initial stage of the secondary arc), which has been little researched in relevant studies. This could increase our fundamental understanding of the secondary arc and long gap alternating-current arc in the free air from a microscopic discharge mechanism. It could help to develop the secondary arc suppression technique and ensure the timely extinction of the secondary arc, and thus contribute to optimizing single-phase auto-reclosing (SPAR). This is of great significance to the safe operation of power transmission lines and enhances the stability of the power system. An experiment platform was established to verify the numerical model. Then the particle density distribution, electric field variation, and time dependence of ion reactions during the discharge were analyzed. The main conclusions are as follows:

(1) The brightness distribution obtained by high-speed cameras of the experimental short-circuit arc was basically consistent with the predicted distribution of electron density, demonstrating that the simulation was effective and supported the subsequent analysis of the plasma interior.

(2) With the short-circuit discharge, the electron density along the ignition line first increased and then decreased, and its distribution was quite different from the general streamer discharge. Over time, the concentration of negative ions rose and then levelled off, and due to the differences of diffusion, convection, and adsorption coefficients between positive ions and negative ions, the changing curves of concentrations of positive ions and negative ions had slight differences despite showing the same trend. Near the end of the simulation time, there was a considerably larger number of charged particles than the initial level, which provided the necessary environmental conditions for subsequent secondary arc generation.

(3) The initial stage of discharge was mainly point discharge in space. The spatial electric field intensity showed an S-shaped upward trend in the discharge process. The end regions were Sssignificantly affected by the high-voltage electrode, whereas the middle area was mainly affected by the particle reaction.

(4) The time correspondence between the detachment reaction and the ion source generated in the short-circuit discharge process was basically consistent, and the detachment reactions were mainly concentrated in the middle area and near the negative electrode. The average recombination reaction rates were consistent with the trend of the detachment reaction rate during the discharge, with differences only in magnitude.

**Author Contributions:** Conceived and designed the numerical model, J.L.; performed the experiments and analyzed the data, J.L., H.Y. and M.J.; wrote the main manuscript text, J.L.; gave important advice on the model, H.L. and G.L.; all authors read and approved the final manuscript.

**Funding:** This research received no external funding.

**Conflicts of Interest:** The authors declare no conflict of interest.

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
