# Peer review of "Numerical Modeling of Space–Time Characteristics of Plasma Initialization in a Secondary Arc"

_energies, doi:10.3390/en12112128_

Round 1

Reviewer 1 Report

The paper presents a numerical model dedicated to the research of plasma initialization in the secondary arc. Using the finite element method software COMSOL, the authors achieved important insight to the plasma process of the secondary arc. The results presented in the paper can be applied in its supression. The simulation results are supported by experiments.

Author Response

I am grateful to you for the valuable suggestions provided. Thank you for your recognition. In the further study, we should take more account of the multi-physics interaction with external environmentin in the simulation and the current and voltage waveforms of the experiment will be analysed. 

Reviewer 2 Report

The paper presents practical and simulation modelling of plasma dynamics for secondary arc phenomena

The paper presents practical and simulation modelling of plasma dynamics for secondary arc phenomena.

The paper is generally well written with a few minor grammatical detected. The work represents some interesting validation of the mechanism and plasma reactions occurring within secondary arc initiations thus provides a relevant contribution to the field.

Some issues:

General

Has the simulation model and procedure been validated using other data/literature for consistency? Quality of some of the images could be improved. Some very minor grammatical errors need to be correct at various places in the paper.

Page 3 Table 1 and lines 82-87 – parameter selection requires more detailed justification and explanation. Must not simply quote numbers need to justify their choice. How will results be affected if different parameters are chosen? Which parameters are most sensitive to the simulation results?

Page 3 lines 97-98 – Is the mesh optimised? No mention of any optimisation process.

Page 4 lines 102-107 – Explanation for choosing the particle source values/parameters and the Gaussian pulse function need to be provided. What happens if parameters are different?

Page 5 line 113ff. - The boundary conditions should be briefly explained to facilitate a better appreciation of the simulation conditions.

Page 5 line 131. How were the L and C values decided? Are these critical for the experiment in terms of arc initiation?

Page 6 lines 141ff. - An explanation of the variability of the practical arc warped nature is required. In repeatable measurements how similar are the photographs? Further comment and explanation of the non-uniformity of the arc diffusion compared to the simulation results should be provided.

Page 12 Conclusions - Discussion of the importance of the work to the field is missing. What are the implications for arcing events and future understanding of arcing events under different scenarios? What are the potential weaknesses or omissions of the model and how can these be overcome?

Author Response

I am grateful to you for the valuable suggestions provided. The corresponding responses are in the file below.

Reviewer 3 Report

In this study, COMSOL multi-physical field coupling analysis software was used to simulate the  short circuit discharge process during the initial stages of a secondary arc fault based on the relevant partial differential equations. The model particularly focused on the complex plasma created by the short circuit discharge, which can increase our fundamental understanding of the secondary arc and  help to optimize the SPAR. And an experiment platform was established to verify the numerical model. Then the particle density distribution, electric field variation, and time dependence of ion  reactions during the discharge were analyzed.

The presentation and quality of work is satisfactory.

Proofread the manuscript for typographical errors.

The results as presented and discussed are satisfactory.

Do shorten the abstract and conclusion.

Author Response

   I am grateful to you for the valuable suggestions provided. Thank you for your recognition. The corresponding responses are in the file below.

Reviewer 4 Report

Motivation and Problem Statement: The first two paragraphs in the introduction (line 29-40) give background on the problem of secondary arcs caused by impermanent (transient) single-phase-to-ground faults in power transmission systems. A few additional sentences summarizing the mechanism of creating secondary arcs would be useful background. As I understand the problem from other references, a fault in a single phase triggers circuit breakers to open the faulted line for a sufficiently long time for the arcing event to pass. After the arcing event ends, the breakers autoclose to restore the line back to an operating condition. Secondary arcing occurs from capacitive and inductive coupling from other line phases after an initial arc generates a conductive plasma in the vicinity of the faulted conductor.  

Additional background would make the paper appeal to a larger audience and also provide a framework for the methodology used to investigate secondary arcing.

Objective: (lines 24 and 25 in the Abstract and lines 53-56 in the Introduction). The authors state the objective of the paper is to model the plasma process in the secondary arc and compare with experimental results. The topic is interesting, but I am less convinced that the experimental results as given in the paper validate the model.

Methodology and Results:

Figure 4 (a) shows a schematic of the experimental test bed and (b) shows pictures of the test bed. A t=0, a conductor short circuits the source through an inductor. The short circuit current ignites the wire which ionizes gases around the conductor. At t=0.1s, the short circuit switch S1 opens and S2 switches on. A high speed camera captures images of the arcing event and results are shown in Figure 5. The arcing images are compared with electron density plots generated by the COMSOL model. Parameters are selected in the model including initial charge distributions. How are these parameters justified and how can they be matched with experimental results? What was the composition of the gas in the lab and are the model parameters for carrier densities justified? What was the humidity? What was the temperature?  We can intuitively relate the brightness of the arc to the density of charge carriers in the plasma, but how can we relate the model to the measurements? Was the model tuned to match the experimental results? It is very difficult to link the experiment with the numerical simulation.

Figure 4(a) shows a current transformer and capacitive voltage divider connected to an oscilloscope to monitor current and voltage. This would be very interesting data to include in the paper. What was the short circuit current waveform? What was the voltage waveform across the conductor during the arcing event? How were the inductance and capacitance values selected in the circuit to model the condition of a typical single-phase-to-ground fault condition? 

The transient electric field at the midpoint of the conductor is shown in Figure 10. It is evident from the plot that the electric field intensity appears to converge to a steady state value over the 0.1 s to 0.15s interval. Is this what is described in lines 208 and 209? The way the sentence reads, it sounds like steady state may be beyond the simulation interval, but the graph appears approach a steady value by 0.2 s. 

Can the authors justify the magnitude of the electric field in Figure 10? As a very rough estimate, the electrodes are 1 m apart and the steady state potential across the electrodes is 600 kV. That would roughly calculate to 6e5 V/m which is three orders of magnitude less than the scale shown in the figure. The steady state value in the simulation appears to be approximately 1.4e8 V/m. Why is this so high?

The legibility of some figures are poor. For example, the color bar scale on the right hand side of Figure 9 is illegible (I assume it is scaled in 10^8 V/m as in Figure 10). Also the labeling of the axes units is crude in many cases where exponentiation is not implemented using superscripts. Overall the quality of figures should be improved. Similar examples appear in the text. Examples: line 85, equations in line 108, and line 113.

Equation (7) uses lower case n to denote carrier concentration while upper case N appears to be preferred notation. Variable notation should be consistent.

Figure 1 legend has a typo in ionization. Also see comments above regarding axes labels.

Lines 94 and 95 describe the Comsol simulation setup. The '.. Bessel curve, rectangle and other geometric entities ..' This was awkward when I first read this sentence. Base objects where used to construct a mesh structure to model the insulator, electrodes, conductor, and space charge region around the conductor. Circular symmetry appears to have been exploited in constructing the geometric configuration. A revised description of the simulation setup is recommended.

In Figure 2, the region around the insulator and conductor is asymmetric with more space above than below. Any particular reason this was selected? Was the volume reasonable for simulation. How was the volume selected? Were the boundaries extended to ensure the that selected volume provides good resolution for the problem?

Table 2 (line 124) shows boundary conditions. In the first line, the partial derivative of the carrier density with respect to radial distance r is set to zero, where zero is bolded to be a vector (I assume). A scalar on the left hand side is equal to a vector on the right hand side? Don't you need to specify a location where the boundary condition applies as well? A unit vector n is included in other boundary conditions and the relations are not clear because of inconsistencies between vector and scalars. The column titled 'Application mode' appears to be the location of the boundary condition. Perhaps this column should be labeled as location.

Author Response

(The authors gave the same response as above.)

Round 2

Reviewer 4 Report

Reviewed your responses to comments and changes have significantly improved the paper. 

This manuscript is a resubmission of an earlier submission. The following is a list of the peer review reports and author responses from that submission.